# Adverse events related to ultrasound-guided regional anesthesia performed by Emergency Physicians: Systematic review protocol

**Sean P. Stickles** [1]*, **Deborah Shipley Kane**[1], **Chadd K. Kraus**[2], **Robert J. Strony** [2], **Enyo A. Ablordeppey**[1], **Michelle M. Doering**[3], **Daniel Theodoro**[1], **Jacques Simon Lee**[4], **Christopher R. Carpenter**[1]

1 Department of Emergency Medicine, Washington University School of Medicine in St. Louis, St. Louis, Missouri, United States of America, 2 Department of Emergency Medicine, Geisinger Health Systems, Danville, Pennsylvania, United States of America, 3 Bernard Becker Medical Library, Washington University School of Medicine in St. Louis, St. Louis, Missouri, United States of America, 4 Department of Family and Community Medicine, Mount Sinai Hospital, Schwartz/Reisman Emergency Centre, Toronto, Ontario, Canada

* spstickles@gmail.com

## Abstract

The use of ultrasound-guided regional anesthesia for pain management has become increasingly prevalent in Emergency Medicine, with studies noting excellent pain control while sparing opioid use. However, the use of ultrasound-guided regional anesthesia may be hampered by concern about risks for patient harm. This systematic review protocol describes our approach to evaluate the incidence of adverse events from the use of ultrasound-guided regional anesthesia by Emergency Physicians as described in the literature. This project will also seek to document the scope of ultrasound-guided regional anesthesia applications being performed in Emergency Medicine literature, and potentially serve as a framework for future systematic reviews evaluating adverse events in Emergency Medicine.

## Introduction

### Rationale and objectives

The use of ultrasound-guided regional anesthesia (USGRA), or ultrasound-guided peripheral nerve blocks, for pain control and painful procedures, has become increasingly prevalent in Emergency Medicine (EM), with a recent survey noting 84% of respondents employing USGRA in their EM training programs 1]. The variety of nerve targets successfully performed by Emergency Physicians (EPs) has expanded over time, and includes both single nerves (e.g. femoral nerve) and fascial planes (e.g. fascia iliaca, serratus anterior plane) [1]. Nearly universally, USGRA has been noted to provide effective pain control and reduce opioid-equivalents administered [2–4]. However, in a 2016 survey, no EM training program employing USGRA reported having a quality assessment program in place [1]. USGRA is not without complications, which can include procedural pain, infection, nerve injury, blood vessel injury, compartment syndrome, allergic reactions, and the potentially fatal local anesthetic systemic toxicity,

relevant data from this study will be made available upon study completion.

**Funding:** The author(s) received no specific funding for this work.

**Competing interests:** The authors have declared that no competing interests exist.

among other adverse events. The primary objective of this study is to systematically review the reported incidence of adverse events related to USGRA performed by EPs.

## Protocol design

This study is a systematic review to summarize randomized controlled trials and observational studies evaluating USGRA performed by EPs and related adverse events. The study protocol adheres to the Preferred Reporting Items for Systematic Reviews and Meta-Analysis (PRISMA) guidelines and recommendations for conducting systematic reviews of adverse events as per Loke, et al. [5, 6]. This review has been registered with PROSPERO (CRD42021241168).

## Study characteristics

This review will include randomized controlled trials and case-control studies investigating USGRA as performed by EPs, evaluating incidence of adverse events related to USGRA as our primary outcome. This review will also consider case series and case studies for inclusion with the purpose of highlighting additional adverse events not reported in more robust trials and describing the scope of USGRA as performed by EPs in the literature. Review articles will be excluded from inclusion. Published studies, studies pending publication, abstracts, conference presentations, and unpublished studies will be considered for inclusion [7, 8].

## Methods

### Eligibility criteria

This review will include studies evaluating participants of all age groups, with subgroup analysis of children (age < 18) and adults (age > 18).

### Interventions

All ultrasound-guided peripheral nerve and fascial plane blocks by EPs the ED will be considered for inclusion. A secondary aim of this study will be to document the wide variety of USGRA applications performed by EPs published in the literature thus far.

### Outcome measures

The primary outcome will be adverse events resulting directly from USGRA procedures performed by EPs. Consideration for adverse events will include descriptions of "adverse event," "adverse effect," "adverse drug reaction," "side effect," "toxic effect," and "complication", as recommended by Loke, et al. [6]. Specific expectant adverse events may include, but will not be limited to, nerve injury, blood vessel injury, local infection, sepsis, allergic reactions, and local anesthetic systemic toxicity (e.g. perioral paresthesias, tachycardia, dizziness, arrhythmia, seizures, hypotension) and/or use of intralipid emulsion infusion. Information provided regarding adverse events in control/comparator groups will also be assessed.

### Types of studies

Randomized controlled trials, observational studies, and case series and reports will be included in analysis of the primary outcome of adverse events related to USGRA performed by EPs (attending physicians, fellows, and residents). For the secondary aim of documenting the variety of ultrasound-guided nerve and plane blocks performed by EPs, randomized controlled trials, cohort studies, case reports, case series, and case-control studies will be considered.

## Information sources

Our search strategy will be performed with the aid of a trained medical librarian (MD) with experience in conducting searches for systematic reviews [9]. Relevant search concepts will include regional anesthesia, nerve block, ultrasound, ultrasound-guided, Emergency Medicine, and Emergency Department. Databases utilized will include Ovid MEDLINE, Cochrane Central Register of Controlled Trials, Scopus, and clinicaltrials.gov. Search will not be limited by language, but non-English manuscripts will be translated into English using Google Translate, which has been demonstrated to be a valid translation program for manuscript review [10]. Each database will be searched from inception to March 2022, and updated as appropriate when preparing submission for publication.

## Study records

**Data management.** Database search results will be exported to and stored in EndNote (Philadelphia, PA) with duplicate entries removed.

**Selection process.** After the initial search result list is compiled, each title and abstract will be reviewed by two reviewers (SPS, DSK) for relevance. Entries potentially meeting inclusion criteria will be reviewed in full-text for final determination of appropriateness for inclusion. Disagreements between the two reviewers regarding inclusion of a study will be settled by a third team member reviewer (EAA). A flowchart will be published to highlight the step-wise results of the selection process.

**Data collection process.** Reviewers will independently extract study characteristics and data using a standardized data collection form. Reviewers will also reach out to study authors as necessary for clarification and to obtain further data when appropriate [11].

## Data items

Extracted data will include lead author, year of publication, study population, number of participants, nerve block performed, control or comparator intervention, anesthetic used, training of person performing the procedure (i.e. attending, fellow, or resident), study primary outcome and result, number (raw and percentage of study group) of adverse events in nerve block group, and types of adverse events in nerve block group, number of adverse events in control group, types of adverse events in control group, how adverse events were captured and managed (if described), and funding source of the study (if described). All collected data will be included in the final manuscript.

## Risk of bias assessment

Risk of bias will be assessed using the Cochrane Risk of Bias version 2 tool for randomized clinical trials, the Newcastle-Ottawa Scale for non-randomized trials, and the Joanna Briggs Institute critical appraisal checklists for case series and case studies [12–15]. Two independent reviewers (CKK, RJS) will apply the Risk of Bias tool to included studies, with discrepancies handled by discussion or a third independent reviewer (DT) if consensus cannot be reached.

## Data synthesis

Due to the expected heterogeneity of results inherent in including all USGRA applications documented in the literature, a meta-analysis is not planned. Instead, data for similar procedures will be compiled and reported together in whole. Strength of available evidence will be evaluated using the Evidence-based Practice Center guidelines [16].

**Subgroup analysis.** Subgroup data by procedure type (e.g. femoral nerve block, ulnar nerve block, serratus plane block, etc.) age group (i.e. adults [over 18 years of age], elderly [over 65 years of age] and pediatrics [under 18 years of age]), analgesic used, and experience levels of sonographers will be reported, although no meta-analysis for these groups is planned.

## Discussion

Providing adequate analgesia for painful procedures is a core skill in EM [17]. While USGRA was initially described in anesthesia literature, the last 15 years has seen a shift in the approach to analgesia in the ED with USGRA becoming more commonplace, particularly amongst academic centers [1, 18]. However, there remain numerous barriers to further implementation of USGRA into the practice of EPs in academic centers as well as community EDs, paramount to them being procedural knowledge, and by extension, safety [1, 19, 20].

This systematic review will quantify the safety of USGRA performed in EDs as evidenced by available published literature. A previous systematic review evaluating regional anesthesia for hip and femur fractures only included one study utilizing ultrasound [2]. To our knowledge, no systematic review exists that has aimed to detail the safety of USGRA by EPs across all target nerves and tissue planes. The value in this is to determine if certain applications may require more training and expertise than others to optimize patient safety and comfort, as well as provider time and liability. It has been suggested that USGRA may become an integral part of EM resident and fellow training, but determining whether all USGRA applications are "created equal" in terms of safety will be necessary to improve clinician knowledge and advance the practice further [21, 22]. This process of considering applicability and safety prior to widespread utilization has been previously experienced with point-of-care ultrasound as a whole, and serves as an ideal example of a medical advancement adopted from another specialty and applied to EM to patient benefit [23].

USGRA presents a means of optimizing care for numerous patient populations in the ED, particularly those with allergies or intolerances to opioid and sedative medications, and elderly patients who may be more prone to adverse reactions to said medications. However, although there are benefits to USGRA, evaluation of its safety profile when performed by EP is pivotal to patient safety and quality. Our systematic review will serve as a valuable asset to inform clinical decision-making, enhance trainee education, and advance clinical research surrounding USGRA by EPs.

## Supporting information

**S1 File. PRISMA checklist.**
(DOC)

**S2 File. Sample search strategy.**
(DOCX)

**S3 File. Extracted data collection form.**
(XLSX)

## Author Contributions

**Conceptualization:** Sean P. Stickles, Christopher R. Carpenter.

**Data curation:** Sean P. Stickles, Deborah Shipley Kane, Michelle M. Doering.

**Formal analysis:** Deborah Shipley Kane, Chadd K. Kraus, Robert J. Strony, Enyo A. Ablordeppey.

**Methodology:** Sean P. Stickles, Chadd K. Kraus, Enyo A. Ablordeppey, Michelle M. Doering, Jacques Simon Lee, Christopher R. Carpenter.

**Project administration:** Sean P. Stickles.

**Resources:** Michelle M. Doering.

**Supervision:** Sean P. Stickles.

**Writing – original draft:** Sean P. Stickles, Deborah Shipley Kane.

**Writing – review & editing:** Sean P. Stickles, Deborah Shipley Kane, Chadd K. Kraus, Robert J. Strony, Enyo A. Ablordeppey, Daniel Theodoro, Jacques Simon Lee, Christopher R. Carpenter.

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
