## [Decision Letter · Decision Letter 0]

11 Mar 2022

PONE-D-21-35950Adverse events related to ultrasound-guided regional anesthesia performed by Emergency Physicians: systematic review protocolPLOS ONE

Dear Dr. Stickles,

Thank you for submitting your manuscript to PLOS ONE. After careful consideration, we feel that it has merit but does not fully meet PLOS ONE’s publication criteria as it currently stands. Therefore, we invite you to submit a revised version of the manuscript that addresses the points raised during the review process.

We look forward to receiving your revised manuscript.

Kind regards,

Amit Bahl

Academic Editor

PLOS ONE

Journal Requirements:

Reviewers' comments:

Reviewer's Responses to Questions

**Comments to the Author**

1. Does the manuscript provide a valid rationale for the proposed study, with clearly identified and justified research questions?

Reviewer #1: Yes

Reviewer #2: Yes

2. Is the protocol technically sound and planned in a manner that will lead to a meaningful outcome and allow testing the stated hypotheses?

Reviewer #1: Yes

Reviewer #2: Yes

3. Is the methodology feasible and described in sufficient detail to allow the work to be replicable?

Reviewer #1: Yes

Reviewer #2: Yes

4. Have the authors described where all data underlying the findings will be made available when the study is complete?

Reviewer #1: No

Reviewer #2: Yes

5. Is the manuscript presented in an intelligible fashion and written in standard English?

Reviewer #1: Yes

Reviewer #2: Yes

6. Review Comments to the Author

You may also provide optional suggestions and comments to authors that they might find helpful in planning their study.

Reviewer #1: The authors present a protocol for a systematic review of existing literatures published on the safety and variety of ultrasound guided nerve blocks performed by emergency physicians. While there are several systematic reviews looking at regional nerve blocks as well as ultrasound guided regional nerve blocks, I would agree with the authors that none have been published looking at the specific topic which they propose here. Their protocol is precise, well written, and feasible. I commend the authors for their efforts and am excited to see this systematic review published.

There are several items that should be addressed prior to publication of this protocol:

1. In the methods section clarity is needed regarding if studies performed outside of the ED will be considered. The interventions section currently states that all ultrasound guided studies will be considered. However, the remainder of your protocol seems to suggest that only ED based studies will be included.

2. According to PRISMA guidelines for systematic review protocol, your section discussing search strategy should include a complete draft search strategy for at least one database including planned limits such that it could be repeated.

3. Should consider adding funding source as a collected data item from the studies.

4. If intended by the authors to allow for open access to their data, there is currently no description of how data will be made available after publication.

5. Consider including your standardized data collection form as a supplementary figure.

Reviewer #2: I believe that this systematic review will provide useful information at a time when emergency provider performed USGRA is rapidly increasing. There is a lot of information regarding safety and efficacy of these procedures in orthopedic and anesthesiology literature, but this same body of research does not exist in emergency medicine.

Eligibility criteria - it may be relevant to include a subgroup analysis of geriatric patients (>65yo), as this is a patient population that oftentimes benefits greatly from USGRA. There is an especially large body of orthopedic literature stating that fascia iliaca compartment blocks/PENG blocks, femoral nerve blocks, etc.) decrease need for breakthrough pain/opioid medications, decrease hospital length of stay and morbidity/mortality, decrease time to post-surgical ambulation, decreased PNA/UTIs/nosocomial infections after surgery. The population that sustains these hip fractures are often >65yo, which may warrant further analysis.

Lines 29-32 state that "USGRA is not without complications, which can include procedural pain, infection, nerve injury, blood vessel injury, allergic reactions, and the potentially fatal local anesthetic systemic toxicity, among other adverse events." There is a lack of substantial evidence regarding USGRA effects on detecting compartment syndrome. This may specifically be an adverse event/sequelae of extremity injuries that is worth investigating as this is oftentimes a concern that orthopedics has regarding administration of these nerve blocks.

7. PLOS authors have the option to publish the peer review history of their article (what does this mean?). If published, this will include your full peer review and any attached files.

Reviewer #1: No

Reviewer #2: No

---

## [Author Response · Author response to Decision Letter 0]

30 Mar 2022

Dear Dr. Bahl and PLoS One Reviewers,

Thank you for your insightful comments on our submission and for your recommendations on improving our work. Overall, we felt your suggestions were helpful and improve the quality of our work. We also appreciate you highlighting some of the formatting particulars that were initially overlooked.

Below, have included our responses in brackets beside the original comments, indicating how we have addressed the concerns in our revised manuscript. 

Additionally, we noticed that one of the authors names (Dr. Lee) had a misspelling in his name on the title page. This has been corrected.

Sean Stickles, MD (corresponding author)

----

Journal Requirements:

https://journals.plos.org/plosone/s/file?id=ba62/PLOSOne_formatting_sample_title_authors_affiliations.pdf -- [This has been corrected.]

--[Thank you for pointing this out, this has been added to the manuscript file itself.]

--[As a protocol paper, we do not have any current data sets to present. However, we have provided clarification that our intent is to publish all available collected data extracted from our review.]

--[All references listed are correct at this time to the best of our knowledge.]

Reviewers' comments:

Reviewer's Responses to Questions

Comments to the Author

1. Does the manuscript provide a valid rationale for the proposed study, with clearly identified and justified research questions?

Reviewer #1: Yes

Reviewer #2: Yes

2. Is the protocol technically sound and planned in a manner that will lead to a meaningful outcome and allow testing the stated hypotheses?

Reviewer #1: Yes

Reviewer #2: Yes

3. Is the methodology feasible and described in sufficient detail to allow the work to be replicable?

Reviewer #1: Yes

Reviewer #2: Yes

4. Have the authors described where all data underlying the findings will be made available when the study is complete?

Reviewer #1: No 

Reviewer #2: Yes

5. Is the manuscript presented in an intelligible fashion and written in standard English?

Reviewer #1: Yes

Reviewer #2: Yes

6. Review Comments to the Author

You may also provide optional suggestions and comments to authors that they might find helpful in planning their study.

Reviewer #1: The authors present a protocol for a systematic review of existing literatures published on the safety and variety of ultrasound guided nerve blocks performed by emergency physicians. While there are several systematic reviews looking at regional nerve blocks as well as ultrasound guided regional nerve blocks, I would agree with the authors that none have been published looking at the specific topic which they propose here. Their protocol is precise, well written, and feasible. I commend the authors for their efforts and am excited to see this systematic review published.

There are several items that should be addressed prior to publication of this protocol:

1. In the methods section clarity is needed regarding if studies performed outside of the ED will be considered. The interventions section currently states that all ultrasound guided studies will be considered. However, the remainder of your protocol seems to suggest that only ED based studies will be included. – [This was clarified.]

2. According to PRISMA guidelines for systematic review protocol, your section discussing search strategy should include a complete draft search strategy for at least one database including planned limits such that it could be repeated. – This is now included.

3. Should consider adding funding source as a collected data item from the studies. –[This is now included.]

4. If intended by the authors to allow for open access to their data, there is currently no description of how data will be made available after publication. –[We anticipate all data collected will be included in a table or tables in the final manuscript. We have added a sentence to the manuscript indicating such.]

5. Consider including your standardized data collection form as a supplementary figure.—[This is now included.]

Reviewer #2: I believe that this systematic review will provide useful information at a time when emergency provider performed USGRA is rapidly increasing. There is a lot of information regarding safety and efficacy of these procedures in orthopedic and anesthesiology literature, but this same body of research does not exist in emergency medicine.

Eligibility criteria - it may be relevant to include a subgroup analysis of geriatric patients (>65yo), as this is a patient population that oftentimes benefits greatly from USGRA. There is an especially large body of orthopedic literature stating that fascia iliaca compartment blocks/PENG blocks, femoral nerve blocks, etc.) decrease need for breakthrough pain/opioid medications, decrease hospital length of stay and morbidity/mortality, decrease time to post-surgical ambulation, decreased PNA/UTIs/nosocomial infections after surgery. The population that sustains these hip fractures are often >65yo, which may warrant further analysis.—[This is now included.]

Lines 29-32 state that "USGRA is not without complications, which can include procedural pain, infection, nerve injury, blood vessel injury, allergic reactions, and the potentially fatal local anesthetic systemic toxicity, among other adverse events." There is a lack of substantial evidence regarding USGRA effects on detecting compartment syndrome. This may specifically be an adverse event/sequelae of extremity injuries that is worth investigating as this is oftentimes a concern that orthopedics has regarding administration of these nerve blocks.—[This is now included.]

7. PLOS authors have the option to publish the peer review history of their article (what does this mean?). If published, this will include your full peer review and any attached files.

Do you want your identity to be public for this peer review? For information about this choice, including consent withdrawal, please see our Privacy Policy.

Reviewer #1: No

Reviewer #2: No

---

## [Editor Report · Decision Letter 1]

9 May 2022

PONE-D-21-35950R1Adverse events related to ultrasound-guided regional anesthesia performed by Emergency Physicians: systematic review protocolPLOS ONE

Dear Dr. Stickles,

Thank you for submitting your manuscript to PLOS ONE. After careful consideration, we feel that it has merit but does not fully meet PLOS ONE’s publication criteria as it currently stands. Therefore, we invite you to submit a revised version of the manuscript that addresses the points raised during the review process.

**Thanks for making the changes requested.**

**Please upload the data collection form.**

We look forward to receiving your revised manuscript.

Kind regards,

Amit Bahl

Academic Editor

PLOS ONE
---

## [Author Response · Author response to Decision Letter 1]

16 May 2022

Data extraction form uploaded and labeled. Sorry for the confusion.

---

## [Editor Report · Decision Letter 2]

26 May 2022

Adverse events related to ultrasound-guided regional anesthesia performed by Emergency Physicians: systematic review protocol

PONE-D-21-35950R2

Dear Dr. Stickles,

We’re pleased to inform you that your manuscript has been judged scientifically suitable for publication and will be formally accepted for publication once it meets all outstanding technical requirements.

Kind regards,

Amit Bahl

Academic Editor

PLOS ONE
---

## [Editor Report · Acceptance letter]

14 Jun 2022

PONE-D-21-35950R2 

Adverse events related to ultrasound-guided regional anesthesia performed by Emergency Physicians: systematic review protocol 

Dear Dr. Stickles:

I'm pleased to inform you that your manuscript has been deemed suitable for publication in PLOS ONE. Congratulations! Your manuscript is now with our production department. 

Kind regards, 

on behalf of

Dr. Amit Bahl 

Academic Editor

PLOS ONE